Remote sensing image analysis and prediction based on improved Pix2Pix model for water environment protection of smart cities

Wang Li 1
Li Wenhao 1
Wang Xiaoyi xiaolizi1983@hotmail.com 2
Xu Jiping xujiping@139.com 1
1 Beijing Laboratory for Intelligent Environmental Protection, School of Artificial Intelligence, Beijing Technology and Business University , Beijing , P.R. China
2 Beijing Institute of Fashion Technology , Beijing , P.R. China
Huang Chenxi
Electronic publication date: 2023 Apr 26
Publication date: 2023
Volume: 9
Electronic Location ID: e1292
Received 2022 Oct 14; Accepted 2023 Feb 23
Copyright: ©2023 Wang et al.
Copyright year: 2023
Copyright holder: Wang et al.
License: This is an open access article distributed under the terms of the Creative Commons Attribution License, which permits unrestricted use, distribution, reproduction and adaptation in any medium and for any purpose provided that it is properly attributed. For attribution, the original author(s), title, publication source (PeerJ Computer Science) and either DOI or URL of the article must be cited.
License URL: https://creativecommons.org/licenses/by/4.0/

Keywords: Prediction, Remote sensing, Image analysis, Pix2Pix model, Water environment, Smart cities, Artificial intelligence, Deep learning, Spatial-temporal data, Neural network

Funding: The Beijing Outstanding Talent Training Grant for Young Top Teams 2018000026833TD01 The National Social Science Foundation of China 19BGL184 This work was supported by the Beijing Outstanding Talent Training Grant for Young Top Teams (2018000026833TD01) and the National Social Science Foundation of China (19BGL184). The funders had no role in study design, data collection and analysis, decision to publish, or preparation of the manuscript.

==============================
Background

As an important part of smart cities, smart water environmental protection has become an important way to solve water environmental pollution problems. It is proposed in this article to develop a water quality remote sensing image analysis and prediction method based on the improved Pix2Pix (3D-GAN) model to overcome the problems associated with water environment prediction of smart cities based on remote sensing image data having low accuracy in predicting image information, as well as being difficult to train.

Methods

Firstly, due to inversion differences and weather conditions, water quality remote sensing images are not perfect, which leads to the creation of time series data that cannot be used directly in prediction modeling. Therefore, a method for preprocessing time series of remote sensing images has been proposed in this article. The original remote sensing image was unified by pixel substitution, the image was repaired by spatial weight matrix, and the time series data was supplemented by linear interpolation. Secondly, in order to enhance the ability of the prediction model to process spatial-temporal data and improve the prediction accuracy of remote sensing images, the convolutional gated recurrent unit network is concatenated with the U-net network as the generator of the improved Pix2Pix model. At the same time, the channel attention mechanism is introduced into the convolutional gated recurrent unit network to enhance the ability of extracting image time series information, and the residual structure is introduced into the downsampling of the U-net network to avoid gradient explosion or disappearance. After that, the remote sensing images of historical moments are superimposed on the channels as labels and sent to the discriminator for adversarial training. The improved Pix2Pix model no longer translates images, but can predict two dimensions of space and one dimension of time, so it is actually a 3D-GAN model. Third, remote sensing image inversion data of chlorophyll-a concentrations in the Taihu Lake basin are used to verify and predict the water environment at future moments.

Results

The results show that the mean value of structural similarity, peak signal-to-noise ratio, cosine similarity, and mutual information between the predicted value of the proposed method and the real remote sensing image is higher than that of existing methods, which indicates that the proposed method is effective in predicting water environment of smart cities.

Introduction

With the expansion of city scale and the acceleration of industrialization, the problem of water environment pollution is becoming more and more serious. As an important part of smart cities, smart water environmental protection has become an important way to solve water environmental pollution problems. Water environment issues can have a huge impact (Isenstein, Kim & Park, 2020; Nowruzi et al., 2021), not only lead to local water shortage (Ibelings et al., 2016), but also affect local economic development. Research shows that economic development is closely related to environmental standards (Bhatti et al., 2022; Li et al., 2021). Water environment pollution is a sudden event, it is very difficult to control once it happens and will cost a lot of human and material resources (Wang et al., 2017). As a result, one of the important tasks of smart water environmental protection is to predict the occurrence of water environment pollution in advance by analyzing monitoring data, and then the relevant departments can take appropriate emergency measures and make early decisions to achieve twice the result with half the effort and prevent the harm caused by the pollution of the water environment.

Currently, there are two main categories of methods for water environment prediction mechanism-driven methods and data-driven methods. Mechanism-driven models use differential or partial differential equations to describe the interactions of influencing factors, considering physical, chemical, and biological processes in aquatic ecosystems (Wang et al., 2013; Wang et al., 2017; Wang et al., 2018a; Woelmer et al., 2022; Yang et al., 2022). However, the mechanism-driven model does not have good generalization performance because of the different algal growth processes in different water bodies.

Based on the type of data used for modeling, data-driven prediction models can be classified into prediction models based on sensor numerical data and remote sensing image data. Sensor numerical data is a certain number of underwater monitoring points set up uniformly in a certain range of water. This method is easy to be affected by weather, and the data volume and coverage are small, making it difficult to achieve simultaneous measurements over large areas of water. Remote sensing image data is obtained through satellite images, which have the characteristics of strong monitoring timeliness and a wide monitoring range. As hyperspectral data, remote sensing images can not only obtain rich spatial information but also have the characteristics of multidimensionality, correlation, nonlinearity and large data volume. Therefore, deep learning can be used to extract deep-seated space–time information from remote sensing images and better predict future remote sensing images.

The prediction models based on sensor numerical data are mainly divided into two categories: mathematical-statistical methods and artificial intelligence methods. Mathematical statistical models are more generalizable than mechanistic models but do not apply to systems with significant nonlinearity (Wang et al., 2020b; Wang et al., 2019a); artificial intelligence models include both traditional machine learning and deep learning methods. Traditional machine learning has shown strong self-learning and self-adaptive capabilities when dealing with nonlinear problems such as water environment pollution (Feki-Sahnoun et al., 2020; Juan Wu et al., 2020; Yajima & Derot, 2018), and is suitable for complex nonlinear systems (Wang et al., 2019b), but still requires manual feature setting and is not suitable for learning large amounts of data (Kashyap et al., 2022; Wang et al., 2019b; Ying, 2022); while deep learning is a type of representation learning, it is capable of learning a higher level of abstract representation of data and automatically extracting deep features from the data (LeCun, Bengio & Hinton, 2015), and the model capability grows exponentially with increasing depth (Dong, Wang & Abbas, 2021). There are several deep learning networks commonly used, including convolutional neural networks (Jin et al., 2022), recurrent neural networks (Li et al., 2019b), graph neural networks (Jia-hui, 2021), and their improved and combined network models (Hwang et al., 2019; Yu et al., 2018), however, the deep learning methods based on sensor data do not take into account the spatial and temporal characteristics of water bodies in an integrated manner.

Prediction models based on remote sensing image data are mostly deep learning-based methods because it involves image processing. According to the input and output types of models, it can be divided into feature-level prediction and pixel-level prediction. Feature-level prediction refers to extracting spatial and temporal features from remote sensing images and then performing numerical prediction based on the features (Wang et al., 2022). Relevant studies include image prediction methods based on convolutional neural networks (CNN) combined with 2-dimensional Gabor filtering (Bhatti et al., 2021); image time series prediction based on long short-term memory (LSTM) (Jin et al., 2022); prediction based on 2DCNN networks superimposed on channels and 3DCNN networks based on remote sensing image prediction methods (Wang et al., 2020a). The feature-level prediction takes remote sensing images as input, which integrates the characteristics of water bodies in time and space, but its prediction results are still numerical data, which cannot reflect the spatial and temporal distribution of water environment pollution.

Pixel-level prediction refers to the prediction of the pixels of the remote sensing image at the future time. Relevant research includes time series prediction of the remote sensing image based on the convolutional long short term memory network (ConvLSTM) (Ma et al., 2021) and convolutional gated recurrent unit (ConvGRU) (He et al., 2022), which uses zero padding in the convolution process, as a result, image edge information prediction is not very accurate; remote sensing image prediction based on generation adversarial network (GAN) and its variants such as Pix2Pix (Li et al., 2019a; Rüttgers et al., 2019), the adversarial learning method of this kind of methods can improve the prediction accuracy, but it is mainly based on the extraction of spatial features; remote sensing image prediction based on 3D-Unet and 3DPatchGAN’s 3D-Pix2Pix (Bihlo, 2021) can simultaneously extract temporal and spatial features, however, gradient explosion or gradient disappearance is likely to occur during the training process due to the complex structure of the model and numerous parameters. Pixel-level prediction takes remote sensing images as input and output, and the prediction results are still images, which can reflect the future temporal and spatial distribution. However, it is easy to have gradient explosion and gradient disappearance, and it is often difficult to extract temporal and spatial features accurately at the same time resulting in low accuracy of image prediction.

The analysis of various water environment prediction methods shows that the pixel-level prediction method based on remote sensing image data can better reflect the spatial and temporal distribution of water environment pollution. Among them, the Pix2Pix model (Isola et al., 2017) is more suitable for processing spatial pixel conversion of images than other pixel level prediction methods, and theoretically can achieve Nash equilibrium and achieve more accurate pixel-level prediction. However, the original Pix2Pix model can only predict a single image based on a single image’s prediction and cannot extract the time series features of the image because the generator only uses Unet network, and the model is not easy to converge during the training process, which leads to increased training difficulty.

Therefore, to address the problem that the existing remote sensing image prediction methods are difficult to effectively make pixel-level prediction of remote sensing images and thus cannot achieve the prediction of spatial and temporal distribution of water environment in the overall waters. Based on the remote sensing images of chlorophyll-a concentration, the pre-processing and pixel-level prediction methods of remote sensing image time series based on the improved Pix2Pix(3D-GAN) model are proposed in this article. The 3D-GAN model achieves pixel-level prediction of remote sensing images at future moments from two dimensions of spatial aspect and one dimension of time, respectively, for a total of three dimensions. On this basis, the prediction of spatial and temporal distribution of water environment pollution in the overall waters was achieved.

According to the prediction results of the spatial and temporal distribution of water environment, the future spatial and temporal distribution of water environment pollution can be effectively analyzed to provide a basis for decision making in water environment management and facilitate relevant departments to take corresponding measures to achieve the purpose of reducing the degree of environmental pollution and treatment costs.

Preliminaries information

The overall research route of this article, the basic model of remote sensing image prediction, and the remote sensing image data set and system information used in this article are introduced in this section.

Research route

An analysis of remote sensing images and a method for water environment prediction based on the 3D-GAN model are proposed in this article. There are three main parts, as shown in Fig. 1.

Figure 1 Research route of remote sensing image analysis and cyanobacteria bloom prediction based on 3D-GAN.

As can be seen from Fig. 1, the main research route of this paper are:

1) The pre-processing methods of remote sensing image time series are studied. Aiming at the problems that the existing raw data of remote sensing images are not perfect and there are problems of non-uniform scale, missing local information of images and unequal sampling time interval, the scale uniformity based on pixel substitution method, image repair based on spatial weight matrix and time series supplement method based on interpolation method are proposed for pre-processing.

2) The remote sensing image prediction method based on the 3D-GAN model are studied. Aiming at the problem of low effectiveness of existing pixel-level prediction methods for remote sensing images, firstly, the ConvGRU model, which is suitable for extracting image time series information, is connected in series with the Unet network of the original Pix2Pix, and the channel attention mechanism is introduced into the ConvGRU to realize the full extraction of feature characteristics of remote sensing images in time dimension and spatial two dimensions, in total three dimensions. Secondly, a residual structure is added to the down-sampling process of the Unet network, thus the network avoids overfitting and improves the pixel-level prediction capability of remote sensing images.

3) The spatial and temporal distribution prediction method of water environment based on remote sensing images are studied. Based on the predicted remote sensing images, the chlorophyll-a concentration at each location in the water can be visualized, and the spatial and temporal distribution of water environment pollution can be judged based on the chlorophyll-a concentration.

Basic model

During this section, the basic models related to the 3D-GAN model presented in this article are discussed.

Original Pix2Pix model

The original Pix2Pix model is based on conditional general adverse nets (cGAN). Initially, Pix2Pix was used to convert image styles. Figure 2 is a structural diagram of the original Pix2Pix model.

Figure 2 Structure diagram of original Pix2Pix model.

As shown in Fig. 2, the original Pix2Pix model is composed of a generator and a discriminator. The generator is composed of a traditional U-net network. The discriminator is composed of PatchGAN, which divides the input image into N × N, and then calculates the probability of each small block. Finally, the average value of these probabilities is taken as the output of the whole. As a result, the amount of computation is reduced and the training speed and convergence speed can be increased. An input image is received by the generator and converted into a predicted image by encoding and decoding, while the discriminator optimizes the parameters of the generator by discriminating between the predicted image and the real image.

ConvGRU model

One of the problems of the original Pix2Pix model is that time series features of images cannot be extracted, while ConvGRU is suitable for time series feature extraction of images. In ConvGRU, the full connection part of GRU is turned into a convolution operation, which preserves the ability of GRU to extract time series as well as the ability to process image data and extract its spatial–temporal characteristics. So ConvGRU is applied to video detection (Wang, Xie & Song, 2018b), gesture recognition (Zhao et al., 2022) and other fields (Qian et al., 2022).

As shown in Fig. 3, ConvGRU has the advantages of both CNN and GRU, extracting spatial features by CNN and temporal features by GRU, thus extracting temporal features in time series at the same time. ConvGRU can be described as (Shi et al., 2017): (1) Rt=σXt∗Wxr+Ht−1∗Whr+br

(2) Zt=σXt∗Wxz+Ht−1∗Whz+bz

(3) H ~t= tanhXt∗Wxh+Rt∘Ht−1∗Whh+bh

(4) Ht=Zt∘Ht−1+1−Zt∘Ht ˜.

Figure 3 ConvGRU structure diagram.

Here, * denotes convolution operation, ∘ denotes Hadamard product, σ denotes sigmoid function, Wx∼ and Wh∼ denotes 2-dimensional convolution kernel, Xt denotes input image at the current time, Rt is the reset gate, Zt is the update gate, Ht is the hidden state at time t, and H ~t is the candidate set.

U-net model

The generator of the original Pix2Pix model is the U-net network, which was first used to solve the problem of medical image segmentation (Ronneberger, Fischer & Brox, 2015). Figure 4 is the structure diagram of the U-net network.

Figure 4 Original U-net model.

As shown in Fig. 4, the left side of the U-net is the down-sampling network for feature extraction, and the right side is the up-sampling network for feature fusion. In feature extraction, image information will be lost and image resolution will be reduced. In feature fusion, the feature map with a larger size obtained by up-sampling through deconvolution lacks edge information. Therefore, edge features can be completed through feature stitching to obtain a complete feature map. It is for these reasons that U-net networks are widely used in the segmentation of images (Li et al., 2018; Li et al., 2022; Yao & Jin, 2022) and generation of images (Kim, Yoo & Jung, 2020; Manjooran et al., 2021).

Remote sensing image data set and system information

The remote sensing image data in this article comes from the Chinese national science and technology infrastructure platform, National Earth System Science Data Center, Lake Basin sub-center. The remote sensing image of Taihu Lake in China is extracted by MODIS satellite, and the remote sensing image data set of chlorophyll-a concentration distribution in Taihu Lake is extracted by the inversion model.

In this article, the data set is divided into a training set and a test set. In each set, 30 consecutive remote sensing images are used as inputs to the 3D-GAN model, and the 31st remote sensing image is used as a prediction output. In the training set, 100 original remote sensing images were selected from March 11, 2010, through December 27, 2010. After processing the data, 293 remote sensing images were obtained. According to the grouping rules, 263 sets of data were obtained as training samples. In the test set, 18 original remote sensing images were selected from October 17, 2011, through December 30, 2011. After processing the data, 75 remote sensing images were obtained. According to the grouping rules, 45 sets of data were obtained as training samples.

The operating environment of this research system is a windows10 system, 11G running memory, and RTX2080ti graphics card, based on Python 3.6.

Methods

Firstly, the original remote sensing image data is imperfect due to inversion differences, weather variations, poor remote sensing communication, etc., resulting in imperfect time series data that cannot be directly used for predictive modeling, so the pre-processing method of remote sensing image time series is proposed in this article. Secondly, in order to achieve pixel-level prediction for remote sensing images, a prediction method for remote sensing image time series based on 3D-GAN model is proposed in this article. Finally, the predicted remote sensing images can reflect the chlorophyll-a concentration at each location in the overall watershed, from which a judgment can be made on the spatial and temporal distribution of water environment pollution.

Preprocessing method of remote sensing image time series

In this article, corresponding preprocessing methods are proposed to solve the problems of imperfect original time series data of remote sensing images, i.e., non-uniform scales, missing local information of images, and unequal sampling time intervals, including:

1) To solve the problem that all image data scales cannot be unified due to the difference of image data scales in the inversion process, the pixel replacement method is adopted to unify the data scales;

2) To solve the problem of missing information and abnormal data due to the weather change in the process of remote sensing image sampling, the spatial weight matrix is used to repair the remote sensing image;

3) To solve the problem of missing remote sensing images at some sampling times due to poor remote sensing communication, i.e., unequal sampling time intervals, the linear interpolation method is used to supplement the time series data.

Data scale of remote sensing images unification based on pixel substitution method

In the actual remote sensing image, the scale of each image data is not uniform due to the inversion difference between different images, as shown in Fig. 5.

The red dotted line part in Fig. 5 represents the data scale of remote sensing images. The data scale of each remote sensing image uses the same nine color grades to represent the nine concentration ranges of chlorophyll-a, but the concentration ranges represented by the same color grade on different images are not the same. Therefore, it is necessary to develop a unified data scale for all remote sensing images and replace the pixel values of all remote sensing images according to the unified data scale.

First, the mean value of the concentration range represented by each color grade on all remote sensing images is calculated to obtain the unified data scale, and the calculation equations are shown in Eqs. (5) and (6) below. After that, the Euclidean distance between the concentration range represented by each color grade in the original image and the concentration range represented by each color grade in the unified data scale is calculated as shown in Eq. (7), and then the pixel value of each color grade in the original image is replaced by the pixel value of the color grade of the unified data scale with the smallest Euclidean distance. (5) LNimax=avg∑n=1kLinmax

(6) LNimin=avg∑n=1kLinmin

(7) LNimax,LNimin=arg minLmax−LNimax2+Lmin−LNimin2.

Here, LNi(max), LNi(min) are the maximum and minimum values of chlorophyll-a at grade i after data scale unification, LNi denotes the concentration of chlorophyll-a at grade i, Linmax and Linmin are the maximum and minimum values of chlorophyll-a at grade i in the n th sheet in the original data set, k is the total number of samples in the data set, avg denotes the mean value operation, L(max), L(min) refers to the maximum and minimum values of each grade on each remote sensing image.

Figure 5 A comparison chart of remote sensing image data scales.

(A) Remote sensing image on March 12; (B) Remote sensing image on March 17.

Remote sensing image repair based on spatial weight matrix

Clouds and abnormal data may cover some areas during remote sensing image sampling, resulting in the lack of data for some remote sensing images, as shown in Fig. 6. Thus, it is necessary to repair the missing area data after unifying the data scale.

Figure 6 Schematic diagram of missing data in some areas of remote sensing images.

(A) Cloud cover remote sensing image (gray); (B) cloud cover remote sensing image (black); (C) data anomaly remote sensing image (white).

The gray area in Fig. 6A represents the missing data caused by cloud cover, the black area in Fig. 6B represents the missing data caused by cloud cover, and the white area in Fig. 6C represents the missing data caused by abnormal data. The repair of the missing area data can be realized by deducing the spatial relationship of the data around the missing area. In describing spatial relationships, some form of distance is usually used as the weight, and the weight is usually the inverse of the distance considering the contribution of the distance to the missing values. As for remote sensing images, the strength of spatial relationships decreases with distance to a stronger extent than linear proportional relationships, so different weight calculations should be used for different spatial relationships.

Therefore, the missing area is repaired by using different weights for different adjacent points based on the spatial weight matrix. The schematic diagram of spatial weight is shown in Fig. 7. When repairing remote sensing images, first determine the contour of the part to be repaired, and then repair from the contour to the middle part until the area is completely repaired.

Figure 7 Spatial weight matrix.

For the first-order neighbor, the weight is the square of the reciprocal of the Euclidean distance, and for the second-order neighbor, the weight is the reciprocal of the Euclidean distance. Equation (8) shows the specific repair method.

In Fig. 7, the black area is the pixel point with missing data, the dark green is defined as the first-order nearest neighbor, the light green is defined as the 2-order nearest neighbor. (8) Pi,j=∑a=1nw1×P1ia,ja+ ∑b=1mw2×P2ib,jbn∗w1+m∗w2.

Here, w1 is the weight of the first-order nearest neighbor, w2 is the weight of the second-order nearest neighbor matrix, P(i, j) is the pixel value of the missing site, P1(ia, ja) is the pixel point of the first-order nearest neighbor, P2(ib, jb) is the pixel point of the second-order nearest neighbor, n is the number of first-order neighboring pixels, and m is the number of second-order nearest neighboring pixels.

Time series data filling based on linear interpolation

For the remote sensing image time series collected for a long period of time, the poor remote sensing communication and other phenomena may occasionally occur during the sampling period, resulting in missing remote sensing images at some sampling moments, i.e., unequal sampling intervals, as shown in Fig. 8.

Figure 8 The sampling time interval of remote sensing images is different.

(A) Remote sensing image of March 11; (B) remote sensing image of March 13; (C) remote sensing image of March 16; (D) remote sensing image of March 17.

In Fig. 8, the sampling intervals are 2 days, 3 days and 1 day, respectively. In order to supplement the missing image to obtain a complete time series dataset with equal sampling intervals, linear interpolation is used in this article to supplement the remote sensing image time series.

As the original data set has a minimum sampling interval of 1 day, the data set is supplemented according to the sampling interval of 1 day. The equation is as follows: (9) Pm+λ=Pm+λPn−Pmn−m.

Here, P(m + λ) represents the pixel value of the day m + λ, λ is the number of days from m,m< λ <n, m , n are the dates, and P(m), P(n) are the pixel values of the two dates already in the original data. The schematic diagram of linear interpolation is shown in Fig. 9.

Figure 9 Schematic diagram of linear interpolation.

As can be seen from Fig. 9, the pixel values of any one of the two days can be obtained by the pixel values of the two days that are not adjacent to each other.

Remote sensing image prediction method based on 3D-GAN model

To address the problems with existing pixel-level prediction methods, namely that it is difficult to fully extract the time series features from an image, the model is prone to gradient explosion and overfitting, the following improvements are made for the original Pix2Pix model:

1) In the generator, the original Pix2Pix model only uses the U-net network, which can not predict the time series of images. In this article, ConvGRU is connected in series with the U-net network to extract features from image time series. On the one hand, the channel attention mechanism is added to the ConvGRU , and the channel attention mechanism convolutional gated recurrent unit network (CAM-ConvGRU) is constructed, which enables the network to learn important information independently and improve its ability of the network to extract image time series features; on the other hand, the residual structure is added to the down-sampling process of U-net network to avoid gradient explosion or over-fitting, which can realize feature extraction and pixel-level prediction of remote sensing image time series.

2) In the discriminator, the original Pix2Pix model inputs the input image into the discriminator as a label, without considering the time correlation. In this article, the image of the historical moment does the data on the channel to do the overlay as the label, and the time series information is incorporated in the criteria to provide the distinction between real and fake images. Figure 10 is a structural diagram of 3D-GAN model.

Figure 10 Structure diagram of 3D-GAN model.

As shown in Fig. 10, CAM-ConvGRU obtains the remote sensing image of the 31st future moment by extracting time series features from the input 30 historical moments images and then inputting them into the Unet network incorporating the residual structure for coding and decoding operations, and the remote sensing image of the future moment is input to the discriminator with the labels obtained by superimposing the input 30 historical moments on the channel, and the parameters of the generator and discriminator are optimized through adversarial training to finally obtain the accurate prediction image.

Generator improvements

The generator in this article is based on the original Pix2Pix model, and the ConvGRU network with a channel attention mechanism added is tandem with the U-net network with a residual structure added in the downsampling process to achieve the prediction of remote sensing images at future moments.

Construction of CAM-ConvGRU

Channel attention mechanisms can not only enable the network to learn the allocation of attention autonomously but also help the network to fuse various important information. The CAM-ConvGRU model is constructed by adding an channel attention mechanism before the gate of ConvGRU, which allows the network to autonomously fuse information in an image as well as improve its ability to extract information from images. The CAM- ConvGRU structure is shown in Fig. 11.

Figure 11 Structure diagram of CAM-ConvGRU.

Here, the red dashed box part represents the attention mechanism added in this article. The schematic diagram of the attention mechanism is shown in Fig. 12.

Figure 12 Schematic diagram of attention mechanism.

As shown in Fig. 12, the attention mechanism in this article is composed of channel attention. The channel attention mechanism can model the dependency between each feature map and adaptively adjust the characteristic response value of each channel. The operation process of the channel attention mechanism can be described as follows: (10) McF=σMLPAvgPoolF∘F.

The input feature map F is globally average pooling to get the features, and the features are input to a multiply-layer perceptron (MLP), and the output of the MLP is multiplied by the activation function to get the weight of each channel, which completes the channel attention construction, where Mc(F) is the feature map output by the channel attention mechanism, MLP is the multiply-layer perceptron, σ is the activation function, ∘ is the Hadamard product, AvgPool is the average pooling, and F is the input feature map.

As a result of introducing the above attention mechanism before ConvGRU enters the activation function of the gating unit, CAM-ConvGRU can be described as follows: First, the input image Xt of this moment and the hidden state Ht−1 of the previous moment are accepted, and the input Rt of the reset gate attention mechanism is obtained after convolution into the channel attention mechanism, and finally the reset gate output rt is obtained after the activation function. (11) Rt=Xt∗Wxr+Ht−1∗Whr+br

(12) rt=σMcRt.

The output of the update gate is similar to the output process of the reset gate, which can be expressed as follows. (13) Zt=Xt∗Wxz+Ht−1∗Whz+bz

(14) zt=σMcZt

(15) H ~t=Xt∗Wxh+Rt∘Ht−1∗Whh+bh

(16) Ht=zt∘Ht−1+1−Zt∘Ht ˜.

Rt is the input of the reset gate attention mechanism, Xt is the current moment input, Ht is the hidden state at moment t, Mc is the attention mechanism, rt is the reset gate output, Zt is the input of the update gate attention mechanism, zt isthe output of the update gate, Ht ˜ is the candidate set, ∗ is the convolution operation, σ is the activation function, ∘ is the Hadamard product, Wx∼ and Wh∼ denotes 2-dimensional convolution kernel.

Construction of U-net network with residual structure

According to U-net, the residual structure is added to the down sampling-layer. The residual structure helps to avoid gradient explosion and gradient disappearance and improves the stability of the network model. The structure diagram of the U-net network integrated with residual structure is shown in Fig. 13.

Figure 13 U-net network structure diagram with residual structure.

As shown in Fig. 13, the U-net network incorporating the residual structure is composed of seven down-sampling layers, seven up-sampling layers, and one output layer. By encoding and decoding the input time series features, remote sensing images are predicted at a future date. The improved down-sampling module is shown in Fig. 14.

Figure 14 Structure diagram of U-net down-sampling module with residual structure.

The red dashed box in Fig. 14 indicates the residual structure added in the down-sampling layer. Each down sampling layer is composed of a conv2d module and a residual module. The conv2d module is composed of a 2-dimensional convolution layer, a batch normalization layer, and an activation function layer. The calculation of the conv2d module is shown in Eq. (17): (17) xout=fBxin∗ω+b.

In Eq. (17), xin is the input time series feature, ω and b represents the weights and bias values of the convolution kernel, * represents the convolution operation, B represents the batch normalization process, f represents LeakyReLu function, xout represents the output after the conv2d module.

The residual module consists of two conv2d modules and one connection layer. In the residual module, the number of convolution kernels of the second conv2d module is twice that of the first conv2d module, and the size of the convolution kernels are 1 ×1 and 3 ×3 respectively, with a step size of 1. The network output size is guaranteed to be the same as the input size after the conv2d module, and the output is superimposed with the input of the residual module after the two conv2d modules, The residual module equation is shown below: (18) y1=fBxout∗ω1+b1

(19) y2=fBy1∗ω2+b2

(20) yout=xout+y2.

Here, yout is the residual module output, xout is the output of the previous conv2d, ω1 and ω2 are the weights, b1 and b2 are the bias values, and y1 and y2 are both intermediate variables. See Fig. 15 for the structure diagram of the up-sampling module.

Figure 15 Structure diagram of U-net up-sampling module.

As shown in Fig. 15, the red dashed box indicates the conv2d module added to the up- sampling layer for feature extraction. The up-sampling module is composed of an Deconvolution layer, a batch normalization layer, an activation function layer, a dropout layer, and a conv2d model with a step size of 1. Dropout layers provide diversity to the network and prevent it from overfitting. Because of the addition of the conv2d module for feature extraction in the up-sampling and the network structure with residuals in the down-sampling, the network is able to generate sharper images and avoid the network degradation caused by the deepening of the network layers.

Improvement of discriminator

In the discriminator network, four conv2d modules comprise PatchGAN. The structure of the discriminator in this article is shown in Fig. 16

Figure 16 Schematic diagram of the discriminator.

As shown in Fig. 16, the discriminator guides the output of the discriminator by using the 30 remote images of historical moments that are input to the generator are superimposed on the channel dimension as labels, ensuring both consistency of labels and providing criteria to distinguish between genuine and fake images while considering temporal features.

Spatial and temporal distribution prediction of water environment based on remote sensing images

After getting the remote sensing image of the future moment, the chlorophyll-a concentration at each pixel position in the remote sensing image is displayed visually, and according to the magnitude of chlorophyll-a concentration, it is possible to make a judgment on the water environment pollution situation at each pixel position, thus realizing the prediction of the spatial and temporal distribution of the water environment pollution. For the locations with serious water environment pollution in the future moment, corresponding measures can be taken in advance to prevent the occurrence of water environment pollution.

Result

Results of preprocessing remote sensing image time series

Results of unified data scale based on pixel substitution method

A unified data scale was determined based on Eqs. (5) and (6), and the unified data scale is shown in Fig. 17. According to the Fig. 17, the pixel value for each color grade in the original remote sensing image is replaced with the pixel value for the corresponding color grade in the unified data scale according to Eq. (7), thus completing the pixel replacement of the unified data scale is completed. A comparison of remote sensing images before and after unifying the data scale is shown in Fig. 18.

Figure 17 Schematic diagram of the unified data scale.

Figure 18 Comparison of remote sensing images before and after scale unification.

(A) Original remote sensing image on Mar. 16; (B) unified remote sensing image on Mar. 16; (C) original remote sensing image on Apr. 29; (D) unified remote sensing image on Apr. 29; (E) original remote sensing image on May. 30; (F) unified remote sensing image on May. 30.

As shown in Fig. 18, the left figure is the original remote sensing image, and the right figure is the remote sensing image after the unified data scale. It can be seen that the color changes of pixels in some areas of the remote sensing image before and after the unification. By unifying the data scale, the same color grade in all remote sensing images represents the same chlorophyll-a concentration range.

Results of remote sensing image repair based on spatial weight matrix

According to Eq. (8) for remote sensing image repair, the results are shown in Fig. 19, which shows a comparison of remote sensing images before and after repair.

Figure 19 Comparison of remote sensing images before and after repair.

(A) Original cloud cover (gray) remote sensing image; (B) repaired remote sensing image of cloud cover (gray) sensing image; (C) original cloud cover (black) remote sensing image; (D) repaired remote sensing image of cloud cover (black); (E) original data anomaly (white) remote sensing image; (F) repaired remote sensing image of data anomaly (white).

As shown in Fig. 19, black and gray areas indicate data loss due to cloud cover, and white areas indicate data loss due to data abnormality. As shown in Fig. 19, the missing part of the data is supplemented by the surrounding pixels. First, determine the outline of the missing part, then fill the outline of the missing part with the above spatial weight matrix calculation rules, and then fill the middle part gradually until the missing part is completely filled.

Results of time series data filling based on linear interpolation

Figure 20 shows the time series of remote sensing images after linear interpolation obtained using Eq. (9).

Figure 20 Time series of remote sensing images after linear interpolation.

(A) Original remote sensingimage on Mar. 13; (B) interpolated remote sensing image on Mar. 14; (C) interpolated remote sensing image on Mar. 15; (D) original remote sensing image on Mar. 16.

Figure 20 depicts the real data in the original data set as March 13 and March 16, and the filling data obtained by linear interpolation as March 14 and March 15. By linear interpolation, the sampling time interval in the data set is set to 1 day.

Remote sensing image prediction results based on improved 3D-GAN model

Parameter setting

A discussion of the 3D-GAN model parameters is presented in this section.

a) Generator parameter settings

The 3D-GAN generator is composed of CAM-ConvGRU and U-net network in series. The parameter settings of CAM-ConvGRU are shown in Table 1. Parameters in Table 1 were derived based on the empirical method and the data format required. The U-net network is divided into the up-sampling network and the down-sampling network. The parameters of the down-sampling network are shown in Table 2.

Table 1 CAM-ConvGRU parameter setting.

Network layer	Parameter setting	Activation function	
Average pooling layer of channel attention	Output size = 1	NA	
Channel attention full connection layer 1	Out features = 8	ReLu	
Channel attention full connection layer 2	Out features = 32	sigmoid	
ConvGRU	Filters = 32, Kernel size = 4	LeakyReLU	

Table 2 U-net down-sampling network parameters.

Network layer	Number of Convolution kernels	Convolution kernel size	Step length	
Down-sampling1	32	3 ×3	1	
Down-sampling2	64	3 ×3	2	
Down-sampling3	128	
Down-sampling4	256	
Down-sampling5	256	
Down-sampling6	256	
Down-sampling7	256	
Down-sampling8	512	

An empirical method was used to determine the number and size of convolution kernels in Table 2. There are seven layers of up-sampling network in U-net, and the parameters are shown in Table 3.

Table 3 U-net up-sampling layer network parameters.

Network layer	Number of Deconvolution kernels	Deconvolution kernel size	Step length	
Up-sampling 1	256	4 ×4	2	
Up-sampling 2	256	
Up-sampling 3	256	
Up-sampling 4	256	
Up-sampling 5	128	
Up-sampling 6	64	
Up-sampling 7	32	
Output layer	3	4 ×4	1	

As shown in Table 3, the number of deconvolution kernels and the kernelsize of the deconvolution are also empirically derived. Step sizes for upsampling and downsampling should be the same.

b) Discriminator parameter setting

The discriminator constructed in this article is the PatchGAN model, and the network parameters are shown in Table 4.

Table 4 PatchGAN network parameters.

Network layer	Number of Convolution kernels	Convolutional kernel size	Step length	
conv2d-1	32	4 ×4	2	
conv2d-2	64	
conv2d-3	128	
conv2d-4	256	
Output layer	1	4 ×4	1	

As shown in Table 4, patchGAN has five network layers. This network uses the remote sensing image of chlorophyll-a concentration at 30 times in history to predict the remote sensing image of chlorophyll-a concentration at 31 times. First, 30 historical remote sensing images of chlorophyll-a concentration are superimposed as video frames, and an Adam optimizer is used to optimize the network with a learning rate of 0.001.

Prediction results of remote sensing images

The test set in this article predicts remote sensing images from November 16, 2011 to December 30, 2011, a period of 45 days. Due to space limitations, only some representative prediction images are displayed (November 22, November 20, December 9, December 20, December 24, and December 28. Figure 21 shows the prediction images for 3D-GAN, which represents the improved Pix2Pix model presented in this article.

Figure 21 Comparison of real image and the predicted image.

(A) Real remote sensing images; (B) Pix2Pix prediction images; (C) channel-Pix2Pix prediction images; (D) 3D-Pix2Pix prediction images; (E) 3D-GAN prediction images.

Figures 21A and 21E represent the real image and the predicted image based on the 3D-GAN model. As can be seen intuitively, the predicted image using the method described in this article is quite similar to the real image.

It is proposed in this article to evaluate the prediction quality of the model more objectively by evaluating structural similarity (SSIM), peak signal-to-noise ratio (PSNR), cosine similarity (cosine), and mutual information. SSIM measures image similarity based on brightness, contrast, and structure; PSNR can reflect the mean square error between two images; cosin can determine the similarity between two images by calculating the cosine distance between the vectors; mutual information describes similarity by calculating the mutual information between two images. In Table 5, the average and maximum values are shown for the similarity between the predicted image and the real image for each prediction model. The more similar the predicted image is to the real image, the closer the predicted result is to the real result. The range of chlorophyll-a concentration in each part can be clearly seen through the predicted image.

Table 5 Similarity comparison of model prediction results (average/maximum).

Models	SSIM	PSNR	Cosin	Mutual Information	
Pix2Pix	0.72193/0.8937	20.64418/29.82518	0.98931/0.999413	1.34135/1.82436	
Channel-Pix2Pix	0.80687/0.88369	21.77778/26.09918	0.99651/0.99907	1.41064/1.65311	
3D-Pix2Pix	0.88765/0.95819	28.67286/34.9271	0.99832/0.99984	1.81485/2.14151	
3D-GAN	0.90112/0.96185	29.9199/36.0489	0.99869/0.99989	1.85518/2.182748	

As can be seen from the last row of Table 5, the prediction results of 3D-GAN proposed in this article maintain good consistency with the real images in terms of brightness, contrast, structure, mean square error, cosine distance, and mutual information.

Prediction results of spatial and temporal distribution of water environment based on remote sensing image

Due to space constraints, Fig. 22 shows only the actual and predicted images of the remotely sensed images acquired on November 22 and December 20.

Figure 22 Spatial and temporal distribution of cyanobacteria bloom on Nov. 22 and Dec. 20

(A) real remote sensing image on Nov. 22; (B) real remote sensing image on Dec. 20; (C) predicted sensing image on Nov. 22; (D) predicted sensing image on Dec. 20.

Taking Fig. 22 as an example, in the overall view of the predicted 45 days, the chlorophyll-a concentration range is higher in the left area of the center of Taihu Lake as well as the left edge part of Taihu Lake, so it is more likely to have water environment pollution, but the chlorophyll-a concentration is generally lower in the right area of the lake. For example, in the predicted image on November 22, the chlorophyll-a concentration was significantly higher in the upper left part of the lake than in other areas; and in the predicted image on December 20, the chlorophyll-a concentration was higher in the left area of the center of Taihu Lake and the upper edge area, while the chlorophyll-a concentration was at a lower level in most of the rest of the area. Therefore, by strengthening the management of the locations with obviously high chlorophyll-a concentration, we can prevent the emergence of water environment pollution and reduce the harm caused by water environment pollution.

Discussion

In order to compare the effectiveness of the improved Pix2Pix model proposed in this article with the existing newer pixel-level prediction methods for remote sensing image prediction, this article’s method (3D-GAN model), the original Pix2Pix model, the Channel-Pix2Pix model with superimposed historical moment image channels, and the 3DU-net as a generator were used to predict remote sensing images of 3D-Pix2Pix model for remote sensing image prediction. Figure 20 and Table 4 illustrate the following:

1) In the original Pix2Pix model, pixel-to-pixel prediction is possible, but it cannot take into account time correlation; therefore, the prediction effect of image time series is general;

2) In Channel-Pix2Pix, time correlation is taken into account, so the prediction effect is better than that of the original Pix2Pix model, but since each image is superimposed on the channel in accordance with time, it is still operating on 2-dimensional data in essence, so the prediction effect is not as good as 3D-Pix2Pix;

3) As 3D-Pix2Pix uses 3D convolution to extract spatiotemporal features, the prediction effect is superior to Channel-Pix2Pix. However, because 3D convolution is not suitable for extracting image time series features, and the method is not easy to converge during training, the prediction effect is inferior to that of The 3D-GAN model;

4) 3D-GAN introduced in this article can better extract spatiotemporal features and prevent overfitting and gradient explosion than the 3D-Pix2Pix model by introducing an attention mechanism;

5) 3D-GAN proposed in this article is used to predict the changes in pixel values of remote sensing images over time. Therefore, this model is not only effective for pixel-level prediction of chlorophyll-a concentration remote sensing images but also applicable to the pixel-level prediction of various remote sensing images theoretically.

Conclusions

The lack of spatial and temporal predictions of water environment of smart cities at present has led this article to propose a method for predicting the spatial–temporal distribution of water environment pollution by proposing the preprocessing method of remote sensing image time series, the prediction method based on the 3D-GAN model, and the prediction method for water environment.

In this article, the time series of remote sensing images are preprocessed first, and a complete remote sensing image data set is constructed. Then, an improved Pix2Pix model (3D-GAN) is developed for the prediction of remote sensing images at the pixel level in the future. Finally, the spatial and temporal distribution of water environment pollution is predicted based on remote sensing images. These three points summarize the innovation in this article:

1) Complete the time series data of remote sensing images

To make the data set of remote sensing image time series more complete, the methods of unifying the data scale of the pixel replacement method, repairing the image with the spatial weight matrix method, and filling the time series with the linear interpolation method are proposed.

2) Improve the effectiveness of pixel-level prediction of remote sensing images

The 3D-GAN model is presented in this article. To enhance the extraction of time series features, attention mechanisms are combined with a ConvGRU network, and residual structure is introduced into the U-net network to prevent overfitting or gradient explosion.

3) Realize the prediction of spatial and temporal distribution of water environment pollution.

In this study, remote sensing images of chlorophyll-a concentration at future moments are obtained by prediction, and the spatial and temporal distribution of water environmental pollution at the pixel level is predicted based on the chlorophyll-a concentration at each location in the remote sensing images.

In this article, only chlorophyll-a concentration is used as a criterion for evaluating water environment pollution, without considering the effects of wind speed and water flow and other influencing factors, and the method for pre-processing remote sensing images in this article is efficient but the principle is relatively simple. Therefore, in the subsequent study, other influencing factors can be considered as the criteria for evaluating and water pollution, and deep learning model can be used to realize the pre-processing of remote sensing images.

Supplemental Information

Supplemental Information 1 Code

Click here for additional data file.

Supplemental Information 2 Data sampled from March 11, 2010 to July 7, 2010

The data were obtained from the remote sensing image data of chlorophyll a concentration from the Lake-Watershed Science SubCenter, National Earth System Science Data Center, National Science & Technology Infrastructure of China, which had inconsistent data scales, data anomalies and different sampling intervals, and the chlorophyll a concentration unit was µg/L.

Click here for additional data file.

Supplemental Information 3 Data sampled from July 19, 2010 to October 3, 2010

The data were obtained from the remote sensing image data of chlorophyll a concentration from the Lake-Watershed Science SubCenter, National Earth System Science Data Center, National Science & Technology Infrastructure of China, which had inconsistent data scales, data anomalies and different sampling intervals, and the chlorophyll a concentration unit was µg/L.

Click here for additional data file.

Supplemental Information 4 Data sampled from October 4, 2010 to December 29, 2010

The data were obtained from the remote sensing image data of chlorophyll a concentration from the Lake-Watershed Science SubCenter, National Earth System Science Data Center, National Science & Technology Infrastructure of China, which had inconsistent data scales, data anomalies and different sampling intervals, and the chlorophyll a concentration unit was µg/L.

Click here for additional data file.

Supplemental Information 5 Experimental result

Click here for additional data file.

Supplemental Information 6 Test dataset

Click here for additional data file.

Supplemental Information 7 The chlorophyll a concentration unit is µg/L, and the sampling interval after data pre-processing is 1 day, dated from August 26, 2010 to September 6, 2010

The data are remote sensing images of chlorophyll a concentration after data scale unification, remote sensing image repair, and time series filling. Remote sensing images of 30 consecutive moments were used as input to the 3D-GAN model.

Click here for additional data file.

Supplemental Information 8 The chlorophyll a concentration unit is µg/L, and the sampling interval after data pre-processing is 1 day, dated from November 21, 2010 to December 28, 2010

The data are remote sensing images of chlorophyll a concentration after data scale unification, remote sensing image repair, and time series filling. Remote sensing images of 30 consecutive moments were used as input to the 3D-GAN model.

Click here for additional data file.

Supplemental Information 9 The chlorophyll a concentration unit is µg/L, and the sampling interval after data pre-processing is 1 day, dated from March 11, 2010 to June 2, 2010

The data are remote sensing images of chlorophyll a concentration after data scale unification, remote sensing image repair, and time series filling. Remote sensing images of 30 consecutive moments were used as input to the 3D-GAN model.

Click here for additional data file.

Supplemental Information 10 The chlorophyll a concentration unit is µg/L, and the sampling interval after data pre-processing is 1 day, dated from June 3, 2010 to August 25, 2010

The data are remote sensing images of chlorophyll a concentration after data scale unification, remote sensing image repair, and time series filling. Remote sensing images of 30 consecutive moments were used as input to the 3D-GAN model.

Click here for additional data file.

Supplemental Information 11 The chlorophyll a concentration unit is µg/L, and the sampling interval after data pre-processing is 1 day, dated from September 7, 2010 to November 20, 2010

The data are remote sensing images of chlorophyll a concentration after data scale unification, remote sensing image repair, and time series filling. Remote sensing images of 30 consecutive moments were used as input to the 3D-GAN model.

Click here for additional data file.

Additional Information and Declarations

Competing Interests

Author Contributions

Data Availability

The authors declare there are no competing interests.

Li Wang conceived and designed the experiments, performed the experiments, analyzed the data, performed the computation work, prepared figures and/or tables, and approved the final draft.

Wenhao Li performed the experiments, analyzed the data, performed the computation work, prepared figures and/or tables, and approved the final draft.

Xiaoyi Wang conceived and designed the experiments, authored or reviewed drafts of the article, and approved the final draft.

Jiping Xu conceived and designed the experiments, authored or reviewed drafts of the article, and approved the final draft.

The following information was supplied regarding data availability:

The raw data is available in the Supplemental Files.

This raw dataset is based on MODIS satellite images to extract the Taihu Lake area, and then de-clouded and geometrically corrected to extract the chlorophyll concentration of Taihu Lake using our research results (inversion model).

This raw dataset is from Lake-Watershed Science SubCenter, National Earth System Science Data Center, National Science & Technology Infrastructure of China (http://gre.geodata.cn).

The training and test files contain input data, and the result folder contains output data.

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
