# Peer review of "Remote sensing image analysis and prediction based on improved Pix2Pix model for water environment protection of smart cities"

_PeerJ Computer Science, doi:10.7717/peerj-cs.1292_

## Round 0.1 · original submission · Major Revisions

Based on the advice received, your manuscript could be reconsidered should you be prepared to incorporate some revisions.

·

Basic reporting

The article accords with the category of remote sensing image analysis and prediction based on deep learning algorithm. It is more interesting to compare the proposed technique to others known in the literature. Before submitting a revision be sure that your material is properly prepared and formatted. If you are unsure, please consult the formatting instructions to authors.The English of the manuscript must be improved before resubmission. It is strongly suggested that the author obtain assistance from a colleague who is well-versed in English.

Experimental design

In the article, lower sampling is used in many places, and down sampling is used in many other places. The same questions also exist for batch standard and batch normalization, which is quite inconsistent.It is confusing that how to calculate the spatial weight matrix when the first-order neighbor or the second-order neighbor is missing. If a large area are all unknown pixels, how to solve the unknown pixel? For example, the continuous loss of data shown in Figure6.

Validity of the findings

Why the parameter of chlorophyll-a directly can determine the eutrophication degree of water body? It is suggested to add some latest references related to image processing, data mining, time series prediction, etc.

Reviewer 2 ·

Basic reporting

1. The authors propose a water environment prediction method based on the Pix2Pix model. The article is in line with the aim and scope of PeerJ Computer Science. However, several points such as English description should be considered to make the article clearer and more interesting to the audience of the journal.
2. The residual structure in Figure 15 is not well reflected.

Experimental design

1. This paper only trains a model with remote sensing images for a period to predict cyanobacterial bloom at the image level, without considering the growth mechanism and dynamics of water and wind.
2. The authors propose that L2 loss is superior to L1 loss due to its fast convergence speed and simple solution, but there is a lack of comparative experiments related to the convergence speed to verify the superiority of L2 loss.

Validity of the findings

The authors summarize their innovations can be effectively controlled by treating on water environment. However, the article lacks relevant experiments to support this conclusion.

Reviewer 3 ·

Basic reporting

1. Authors propose a water quality remote sensing image analysis and prediction method based on the improved Pix2Pix model for water environment protection of smart cities. The approach is interesting and has certain merits. My main objection is that the manuscript is a little difficult to read and understand, mainly because of poor English. A substantial revision in this respect would be beneficiary.
2. The article is too long and needs to be streamlined. Some of the existing studies need not be detailed in the paper (i.e. original Pix2Pix model, ConvLSTM, etc.).

Experimental design

no comment

Validity of the findings

1. The title of this article is water environment protection of smart city, but the main content of the text is Cyanobacterial bloom. It is suggested to remove the statement of Cyanobacterial bloom and change it to water environment assessment or prediction.
2. The test data of remote sensing images are from November and December, but the period of training data is not given in the paper. It is uncertain whether the growth of water environment in winter can be accurately predicted by using the training data during the other water environment period.
3. The figure 13 does not give the concept of the dotted lines and gray boxes. The residual structure in Figure 13 is not well reflected.

Reviewer 4 ·

Basic reporting

The paper may be regarded for publishing after minor editing.
1. The English of the manuscript must be improved before resubmission. It is strongly suggested that the author obtain assistance from a colleague who is well-versed in English.
2. The article accords with the category of remote sensing image analysis and prediction based on deep learning algorithm. The approach is interesting and has certain merits.
3. In the article, lower sampling is used in many places, and down sampling is used in many other places. The same questions also exist for batch standard and batch normalization, which is quite inconsistent.

Experimental design

1. The test data of remote sensing images are from November and December, but the period of training data is not given in the paper. It is uncertain whether the growth of water environment in winter can be accurately predicted by using the training data during the other water environment period.
2. In addition, there are several other questions in the paper: What is the compensation basis for cloud blocking or data loss? Is the compensation true? In Figure 7, only the solution method for the center pixel is given. If a large area are all unknown pixels, take an unknown pixel as an example, and only a few of the 8 and 16 neighborhoods are determined values, how to solve the unkown pixel?
3. What do the dotted lines and gray boxes in Figure 13 mean? The figure does not give the concept of residual structure.
4. The residual structure in Figure 15 is not well reflected.
5. Figure 19 should give the basis of compensation results for cloud occlusion and missing parts.
6. Table 7 lists some evaluation criteria. Can we directly use the subtraction of pixel values for both images to calculate the difference to judge the accuracy of prediction?

Validity of the findings

1. It is confusing that how to calculate the spatial weight matrix when the first-order neighbor or the second-order neighbor is missing. If a large area are all unknown pixels, how to solve the unknown pixel? For example, the continuous loss of data shown in Figure 6.
2. Why the parameter of chlorophyll-a directly can determine the eutrophication degree of water body?
3. Is the distribution of cyanobacterial bloom related to other meteorological conditions, such as whether? Will the cyanobacterial bloom move under the action of wind and water flow?

---

## Round 0.2 · Minor Revisions

Based on the advice received, your manuscript could be reconsidered should you be prepared to incorporate some revisions.

·

Basic reporting

For me, this article meets the publishing requirements

Experimental design

For me, this article meets the publishing requirements

Validity of the findings

For me, this article meets the publishing requirements

Reviewer 2 ·

Basic reporting

The author has answered all my questions. But there are still some mistakes in the article that need to be corrected. The detail comments are listed as follows.
1.The language of this manuscript needs to be further improved. Some descriptions of the article are too repetitive, and the language should be more concise and accurate.
2.Professional words should be expressed consistently in this article,such as the word 'chlorophyll-a'. Sometimes it's 'chlorophyll-a' and sometimes it's 'chlorophyll a'. They should be unified.
3.Some future research directions should be discussed in Conclusion.

Experimental design

no comment

Validity of the findings

no comment

Reviewer 3 ·

Basic reporting

no comment

Experimental design

no comment

Validity of the findings

no comment

Additional comments

no comment

Reviewer 4 ·

Basic reporting

The title of this article is water environment protection of smart city, but the main content of the text is Cyanobacterial bloom. It is suggested to remove the statement of Cyanobacterial bloom and change it to water environment assessment or prediction.

Experimental design

The subject and scope of the journal are computer science. The contributions should focus on the preprocessing and prediction methods of remote sensing images. The adjustment of traditional water quality standards is introduced too much and should be simplified. Therefore, both Table 1 and Table 2 should be deleted and described in words.

Validity of the findings

In the result part, the classification of eutrophication is too detailed, which is not conducive to improving the efficiency of subsequent decision-making. It is recommended to simplify the classification.

---

## Round 0.3 · accepted · Accept

Your manuscript is now ready to be accepted.

Reviewer 4 ·

Basic reporting

English description is basically clear.

Experimental design

The experimental design basically meets the requirements.

Validity of the findings

The research in this paper has certain contributions.